# CRISPR-Cas9 induces large structural variants at on-target and off-target sites in vivo that segregate across generations

Ida Höijer [1✉], Anastasia Emmanouilidou[1,2], Rebecka Östlund [1], Robin van Schendel [3], Selma Bozorgpana[1], Marcel Tijsterman [3], Lars Feuk [1], Ulf Gyllensten [1], Marcel den Hoed [1,2,4] & Adam Ameur [1,4✉]

CRISPR-Cas9 genome editing has potential to cure diseases without current treatments, but therapies must be safe. Here we show that CRISPR-Cas9 editing can introduce unintended mutations in vivo, which are passed on to the next generation. By editing fertilized zebrafish eggs using four guide RNAs selected for off-target activity in vitro, followed by long-read sequencing of DNA from >1100 larvae, juvenile and adult fish across two generations, we find that structural variants (SVs), i.e., insertions and deletions ≥50 bp, represent 6% of editing outcomes in founder larvae. These SVs occur both at on-target and off-target sites. Our results also illustrate that adult founder zebrafish are mosaic in their germ cells, and that 26% of their offspring carries an off-target mutation and 9% an SV. Hence, pre-testing for off-target activity and SVs using patient material is advisable in clinical applications, to reduce the risk of unanticipated effects with potentially large implications.

[1] Science for Life Laboratory, Department of Immunology, Genetics and Pathology, Uppsala University, Uppsala, Sweden. [2] The Beijer laboratory and Department of Immunology, Genetics and Pathology, Uppsala University, Uppsala, Sweden. [3] Department of Human Genetics, Leiden University Medical Center, Leiden, The Netherlands. [4] These authors contributed equally: Marcel den Hoed, Adam Ameur. ✉email: ida.hoijer@igp.uu.se; adam.ameur@igp.uu.se

Genome editing using the CRISPR-Cas9 system has become an indispensable tool across many areas of biomedical research, and holds promise to revolutionize the treatment of genetic disorders[1–4]. However, the use of CRISPR-Cas9, in particular for human germline gene editing, has raised ethical questions that need careful consideration. One major aspect of attention is unintended mutations, caused by CRISPR-Cas9, at locations in the genome other than the targeted site[5,6]. Such *off-target* mutations can have serious consequences as they might disrupt the function or regulation of non-targeted genes. In addition, larger structural changes of the genome sequence, occurring at the intended *on-target* editing site, are another cause of concern.

Undesired outcomes of CRISPR-Cas9 genome editing have been the subject of many investigations. The conclusions from these studies have been somewhat conflicting, with adverse effects of CRISPR-Cas9, i.e., larger on-target structural variants (SVs) and off-target mutations, being reported in some cases[5–8] but not in others[9,10]. These discrepancies can, at least partly, be explained by differences in experimental factors such as the Cas9 concentration, delivery method, or specific properties of the cells being investigated. In other cases, limitations of the experimental setup or the genomics technologies used to interrogate the editing sites, could hinder the discovery of CRISPR-Cas9-induced events. Moreover, the adverse CRISPR-Cas9 effects may be rare and only occur in a small fraction of the edited samples. Therefore, in order to conclusively determine the effects of CRISPR-Cas9 and their long-term consequences in vivo, a large number of samples needs to be followed through development and over generations, using a sensitive method for genome analysis.

Validation of genome editing is often performed using short-read or Sanger sequencing. While such methods are capable of detecting small insertion and deletion events, which are the most common outcomes of CRISPR-Cas9 genome editing, they may fail to detect larger genome aberrations. Long-read sequencing technologies suffer less from these limitations. In a pioneering study by Kosicki et al., large deletions and complex rearrangements were shown to exist at the on-target site of genome-edited cells, through a combination of long-range PCR and long-read sequencing[11]. Following this study, Cas9-induced SVs have been detected in vivo at the on-target site[12–14]. Recently, reports have emerged describing other types of complex genome rearrangements at the on-target site, including segmental or whole chromosome deletions[15–20] as well as chromothripsis[21].

Large SVs and complex genome aberrations induced by CRISPR-Cas9 have mainly been observed at the on-target sites, although Zuccaro et al. recently reported such effects also at off-target sites through experiments in human embryos[20]. Genome aberrations in chromosomal regions or genes not intended or monitored for editing could lead to unpredictable functional consequences, and are therefore arguably more worrying than unintended alterations at the on-target site. To examine whether SVs at off-target sites are a cause for concern, their genomic locations first need to be established. The off-target locations can be predicted by computational tools[22–26], but a more reliable approach is to experimentally determine the Cas9 off-target activity in vitro using a sequencing assay[27–32]. For this purpose, we recently developed Nano-OTS, a long-read sequencing assay based on nanopore sequencing[33]. The Nano-OTS method does not suffer from amplification bias, and reliably identifies off-target sites, even in repetitive and complex regions of the genome.

In this work, we aim to gain a better understanding of unintended CRISPR-Cas9 genome editing outcomes at on- and off-target sites in vivo, and in particular SVs that may escape detection by short-read or Sanger sequencing. To accomplish this, we use DNA from a large number of CRISPR-Cas9-edited zebrafish (*Danio rerio*) and their offspring, and examine the on-target and off-target sites with long-read sequencing. Though our study is performed in zebrafish, we expect that the findings can be extrapolated also to other vertebrate animals, including humans, provided that the editing experiments are performed under similar conditions. Finally, we propose a strategy for detection and validation of CRISPR-Cas9 genome editing outcomes using long-read sequencing technology, which we believe will represent an important step towards reducing the risk of adverse effects of CRISPR-Cas9 in clinical applications.

## Results

**Detection of Cas9 off-target cleavage sites in zebrafish DNA.** The aim of this study is to investigate the prevalence and distribution of different types of mutations post CRISPR-Cas9 editing at the intended target site (on-target) as well as at in vitro-established off-target sites. To select gRNAs for our experiments, we pre-screened 23 gRNAs with high in vivo on-target efficiency. These gRNAs target zebrafish orthologues of human genes in loci identified by genome-wide association studies (GWAS) for cardiometabolic risk factors and diseases[34–36] and have been used to examine the role of the candidate genes in cardiometabolic disorders[37,38]. We first used genome-wide Nano-OTS[33] to identify off-target Cas9 cleavage activity in vitro for all 23 gRNAs (Supplementary Tables S1, S2). The four gRNAs with the highest number of in vitro-detected off-target sites, and with at least one of these located within a gene, were selected for further experiments. The four selected gRNAs target early exons of *ldlra, nbeal2, sh2b3,* and *ywhaqa*. Nano-OTS identified five off-target sites for the *ldlra* gRNA (three intronic), 13 for the *nbeal2* gRNA (seven intronic; two exonic), two for the *sh2b3* gRNA (one intronic) and seven for the *ywhaqa* gRNA (one intronic; six exonic) (Fig. 1). The off-target sites had between two and seven mismatches with the gRNA sequence, including mismatches at the PAM site (*nbeal2* off-target 5 and *sh2b3* off-target 1). Further investigation of the off-target sites with mismatches in the PAM sequence revealed an adjacent PAM site located 1 bp downstream of the intended PAM site, with the mismatched base pair being the first nucleotide of the NGG PAM site motif.

**CRISPR-Cas9 genome editing and crossing of founders.** CRISPR-Cas9 editing experiments were set up as outlined in Fig. 2, using the four gRNAs targeting *ldlra, nbeal2, sh2b3,* and *ywhaqa*. To this end, fertilized eggs were microinjected with ribonucleoproteins (RNPs) at the single-cell stage, while uninjected eggs from the same crossing were used as controls. Microinjected RNPs typically result in >90% editing efficiency and have become a method-of-choice for genome editing in functional studies in zebrafish[39]. Samples were collected for analysis at the larval stage (5 or 10 days post-fertilization) as well as when founder fish reached adulthood (3 months). Next, we crossed randomly selected pairs of adult F0 fish to obtain an F1 generation of edited zebrafish. In the F1 generation, we collected samples at the larval stage as well as from juvenile fish (2 months). Three replicate CRISPR-Cas9 genome editing and crossing experiments were performed consecutively, using fertilized eggs from different parents of the same zebrafish line (ABs). A complete list of all examined zebrafish samples is provided in Supplementary Table S3. Since the genome editing was successful in all replicate experiments, with an on-target editing efficiency of at least 84%, all samples collected at the same developmental stage and edited with the same gRNA were jointly analyzed.

**CRISPR-Cas9 induces editing at on- and off-target sites.** To investigate the types of Cas9-induced mutations at on- and off-target sites, we constructed large amplicons (2.6–7.7 kb) spanning the Cas9 cleavage sites in samples from edited zebrafish, as well as in uninjected controls (Supplementary Table S4). The PCR products were

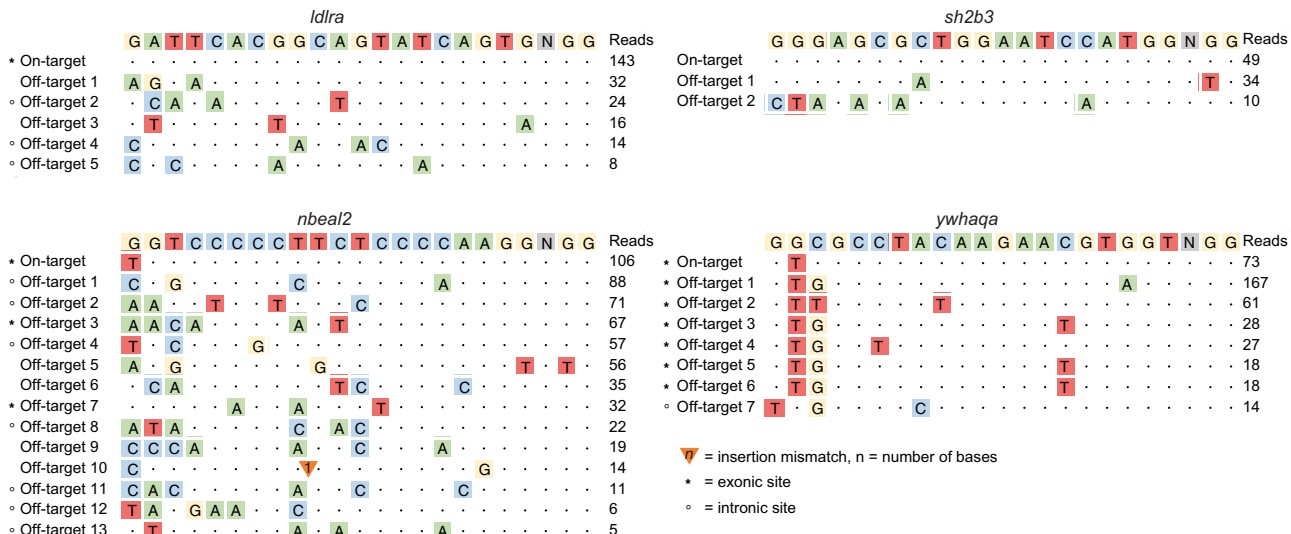

**Fig. 1 Predicted off-target sites of four guide RNAs for zebrafish genome editing.** The diagrams show Cas9 cleavage sites detected in vitro by Nano-OTS for the four gRNAs targeting *ldlra*, *nbeal2*, *sh2b3*, and *ywhaqa*. The sequence at the top of each diagram displays the gRNA sequence and PAM site (NGG). The rows below show the on-target site as well as the identified off-target sites. Colored letters correspond to single-nucleotide mismatches between the target site and the GRCz11 genome. Triangles are used to mark insertion mismatches, where nucleotides need to be inserted to match the reference genome. Asterisks and circles mark off-target sites located within exonic and intronic regions, respectively. The column to the right shows the number of reads in the Nano-OTS analysis for each target site.

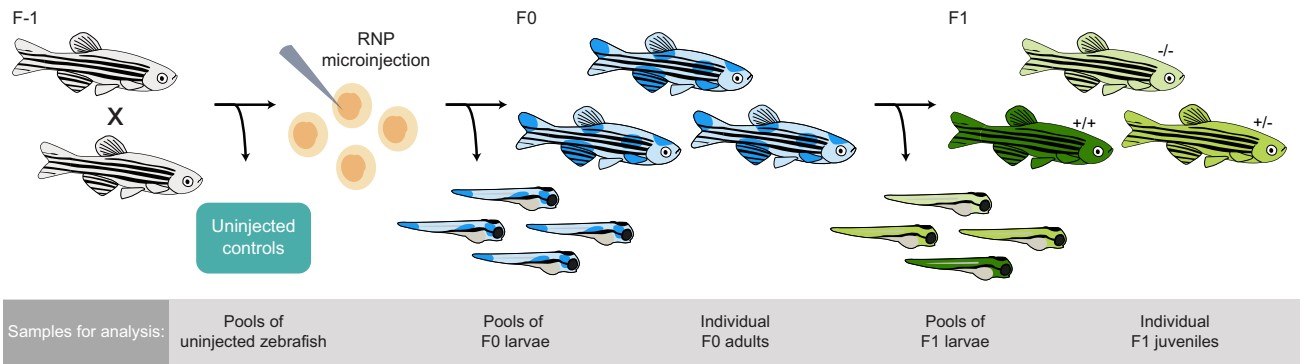

**Fig. 2 Overview of CRISPR-Cas9 genome editing in zebrafish.** Genome editing was performed in fertilized eggs by microinjection of ribonucleoprotein (RNP) at the single-cell stage. The genome editing experiment results in mosaic heterozygous mutants, mosaic homozygous mutants, or unaffected homozygotes (together referred to as founders). A number of F0 embryos were not injected and used as controls. F1 generation zebrafish were generated by in-crossing randomly selected pairs of adult founders. The offspring of these crossings have stable genotypes with 0 (−/−), 1 (+/−) or 2 (+/+) mutated alleles. Samples were collected for analysis at different stages of the experiment as described in the gray box.

sequenced using the PacBio Sequel system, to obtain long and highly accurate (>QV20) reads. Detecting and quantifying genome editing outcomes from the resulting PacBio reads was performed using the software SIQ ("Methods"). To filter out false positives, which for example could occur due to alignment difficulties in homopolymer regions, all events detected in an uninjected control sample were removed from further analyses. On-target editing efficiencies were then calculated based on the remaining insertion or deletion mutations in the pools of founder larvae. This resulted in 92.6% on-target editing efficiency for *ldlra*; 96.7% for *nbeal2*; 92.6% for *sh2b3*; and 93.6% for *ywhaqa* (Fig. 3a). In addition, we identified Cas9 activity at three off-target sites; *sh2b3* off-target 1 (editing efficiency 1.8%); *ywhaqa* off-target 1 (2.4%); and *ywhaqa* off-target 2 (6.3%) (Fig. 3b).

### Founder fish are highly mosaic in somatic and germ cells
We next examined the on-target and the three in vivo-confirmed off-target sites in 26 adult founders at three months of age, edited either for *sh2b3* (*n* = 11) or *ywhaqa* (*n* = 15) at the single cell stage.

18 of 26 F0 fish (69.2%) showed on-target editing (Fig. 3c, d). Many distinct insertion and deletion events, as well as more complex combinations of insertions and deletions, were observed in any single individual, consistent with mosaicism of genome editing outcomes at the on-target site (Fig. 4). To facilitate the downstream analyses, every event that consisted of a combined insertion and deletion was counted either as an insertion or a deletion, depending on which of the two sub-events that involved the highest number of nucleotides. By further examining CRISPR-Cas9-induced mutations in individual F1 juvenile fish and pooled larvae from the same founder parents, we noticed that up to six unique alleles were passed on from a single F0 breeding pair (Supplementary Tables S5–S10). Since at most four alleles are expected at any given locus, this observation is consistent with mosaicism in the founders' germ cells. Six founder fish (23%) displayed off-target genome editing in at least 10% of DNA molecules, with the highest proportion (50.4%) observed in founder individual #10 at *ywhaqa* off-target 2 (Fig. 3d).

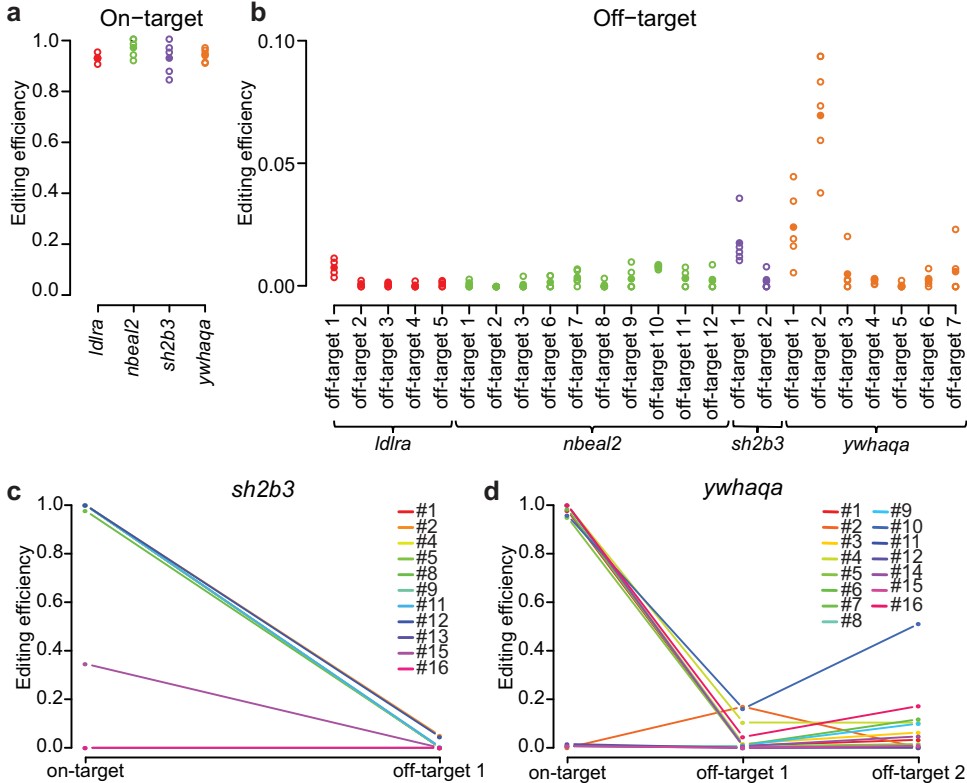

**Fig. 3 CRISPR-Cas9 editing efficiencies in pooled founder larvae and individual fish. a** On-target CRISPR-Cas9 editing efficiencies in pools of founder larvae for the four gRNAs *ldlra*, *nbeal2*, *sh2b3*, and *ywhaqa*. Each circle corresponds to the editing efficiency in a specific larvae pool, and the average values are indicated by solid circles. **b** Off-target CRISPR-Cas9 editing efficiencies in pools of founder larvae. **c**, **d** Line graphs visualizing the CRISPR-Cas9 editing efficiencies at on- and off-target sites in individual adult founders. Each colored line shows the editing efficiencies at the on- and off-target site(s) in one individual. Source data are provided as a Source Data file.

**Large structural variants at on- and off-target sites**. To determine the size distribution of genome editing events induced by CRISPR-Cas9 in vivo, we focused on pools of founder larvae. A total of 595 larvae were analyzed in 20 pools, thereby giving a comprehensive view of the different alleles introduced by CRISPR-Cas9 at an early developmental stage. The on-target events ranged from 4.8 kb deletions to 1.4 kb insertions (Fig. 5a–c). Although the majority of events were small insertions and deletions, 7% represent SVs of ≥50 bases. A similar fraction of SVs (4%) was seen at the three off-target sites for *sh2b3* and *ywhaqa*, even though the lower degree of off-target editing results in a diagram with fewer data points (Fig. 5d–f). When considering on-target and off-target sites combined, the fraction of SVs was 6%. Large SVs were detected not only in founder larvae, but also in founder adults. Strikingly, one 903 bp deletion at an off-target site completely removes an exon of a gene that was not intended to be targeted in the experiment (i.e., *ywhaqb*, Fig. 5g).

**Off-target mutations and SVs can be passed to the F1 generation**. We next compared the frequency of genome editing events over developmental stages and generations. The proportion of edited alleles was higher in the F1 generation, where all 46 juvenile individuals were completely edited, as compared to the F0 generation where eight fish showed little or no editing (Fig. 6a). Structural variants are also more abundant in the F1 generation (Fig. 6b). Four of the 46 juvenile F1 individuals (9%) were hetero- or homozygous for an on-target SV. Editing events at the three off-target sites showed a similar pattern, with 12 of 46 F1 fish (26%) displaying editing in at least 20% of the reads for at

least one off-target site i.e., representing hetero- and homozygous individuals at that locus. Direct comparisons to F0 founder individuals are challenging, due to their mosaic nature which implies that they may carry editing events at variable frequencies. However, six of 26 adult F0 fish (23%) displayed off-target editing at a 10% level, which can be seen as a threshold for moderate off-target activity in the founder generation (Fig. 6c). No SVs were detected at off-target sites in the F1 generation (Fig. 6d).

**Validation of unintended CRISPR-Cas9 editing in F1 fish**. As mentioned above, we identified four juvenile F1 fish with a SV at an on-target site, and 12 F1 individuals with smaller off-target mutations. Our experimental setup enabled us to search for the same events in pools of F1 larvae from the same parents, as well as in other F1 siblings. This way, we were able to verify that all unintended on-target and off-target mutations indeed exist in related larvae and juvenile fish (Supplementary Table S11). Figure 7a, b shows the results for two F1 individuals with large on-target SVs. A 1053 bp deletion, removing a big fraction of the targeted exon of *sh2b3* was found in a juvenile F1 fish as well as in pooled F1 larvae (Fig. 7a, c). Furthermore, a large 292 bp insertion in the targeted exon of *ywhaqa*, observed in three juvenile F1 fish, was also detected in pooled F1 larvae (Fig. 7b, d). Unexpectedly, for seven of the 46 F1 individuals (15%), >98% of the reads support only one specific editing event (Supplementary Table S12). These fish could be homozygous for the edited locus[40], carry a large deletion on the other chromosome, or, alternatively, a different allele exists that was not detected due to allelic dropout. Our results thus confirm that large SVs and off-target mutations are present in the F1 generation,

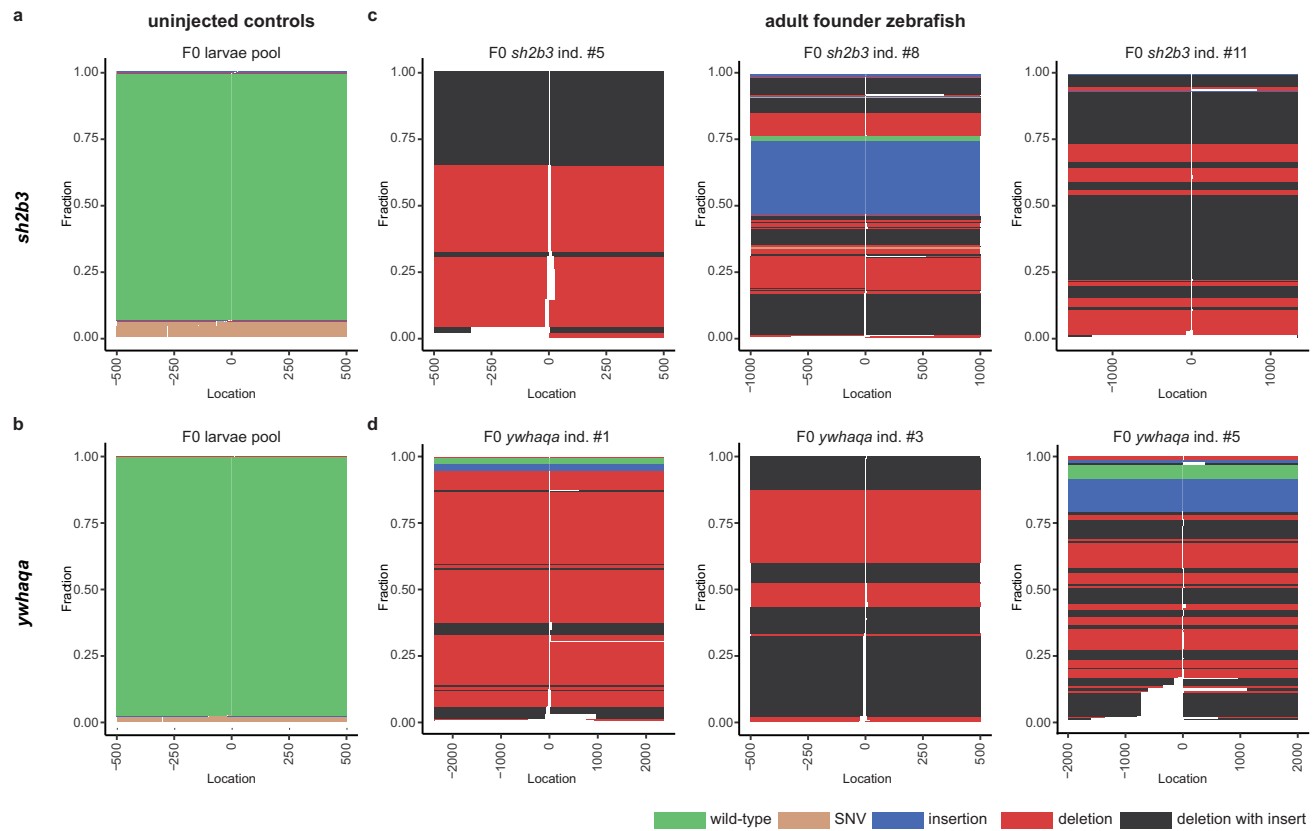

**Fig. 4 Somatic mosaicism of CRISPR-Cas9 editing events in founder fish.** Schematic view of sequencing reads and editing events at the on-target sites for *sh2b3* and *ywhaqa*, produced by the SIQ software. In each plot, the *y*-axis displays the fraction of different editing events, and the *x*-axis their genomic positions. The zero coordinate indicates the Cas9-cleavage site for *sh2b3* and *ywhaqa*. Control samples from uninjected zebrafish displaying the *sh2b3* (**a**) and *ywhaqa* (**b**) targeted sites. Only wild-type alleles and sequences containing SNVs are detected in these controls. SIQ results from three individual F0 founder fish edited for *sh2b3* (**c**) or *ywhaqa* (**d**). In the CRISPR-Cas9 edited F0 individuals, a large number of distinct insertions, deletions, as well as combinations of insertions and deletions are reported. This indicates a high degree of somatic mosaicism of CRISPR-Cas9 editing events in the F0 individuals. Source data are provided as a Source Data file.

but additional experiments are required to understand the underlying cause of the homozygosity observed in several of the F1 individuals.

**Whole genome long-read re-sequencing of edited zebrafish.** Two of the juvenile F1 fish were investigated in detail using nanopore-based whole genome sequencing (WGS). These F1 individuals represent the offspring from two different *ywhaqa* founder pairs, and were selected for WGS based on the skewed on-target allele distribution from long-amplicon sequencing. Individual #5 carried a 21 bp deletion in 97.8% of reads but also a rare 6 bp deletion, while individual #17 was dominated by a 3 bp deletion (in 99.7% of the reads) and was therefore considered homozygous, although a 5 bp deletion was present in a few reads. Nanopore WGS confirmed both individuals to be heterozygous (Supplementary Fig. S1), without large-scale copy number variations (CNVs) in any of the samples (Supplementary Fig. S2). Through SV calling, a heterozygous repeat insertion of size 65–66 bp was detected in both individuals, consisting of a dinucleotide repeat with 32–33 AT-units (Supplementary Tables S13, S14). In both individuals, the repeat coincides with the rare CRISPR-induced allele reported by long-amplicon sequencing, and multiple amplicon reads terminate at the AT-repeat locus (Supplementary Fig. S1). Since the reads that terminate at the repeat are generated from intact SMRTbell molecules sequenced in circular consensus (CCS) mode, and because repeats of much longer size can be examined through amplification-free PacBio

sequencing[41], the allelic dropout in the two examined F1 individuals is likely not explained by errors during PacBio sequencing, but rather by difficulties to amplify dinucleotide repeat stretches by long-range PCR. While we did not determine whether the same explanation holds true for other F1 fish, our results highlight the importance of validating seemingly homozygous samples with an orthogonal method to correctly determine all edited alleles.

## Discussion

In this study, we used long-read sequencing to examine on- and off-target genome editing outcomes, across multiple stages of development and across generations. Genome editing was accomplished using standard routines for germline editing at the single-cell stage, using gRNAs that have recently been used for functional studies of cardiometabolic diseases in zebrafish model systems[37,38]. This revealed insertions and deletions of sizes up to several kilobases, at on- and off-target sites, with a high degree of individual-level variation in genome editing outcomes both in the F0 and F1 generations. A major advantage of our experimental setup is that editing events detected in the F1 generation could be directly verified in siblings from the same parents. In this way, we were able to validate all off-target mutations and larger SVs in the F1 generations. At the same time, we confirmed the absence of events in uninjected controls and in F1 fish from different founders. We can therefore conclude that no false positives were

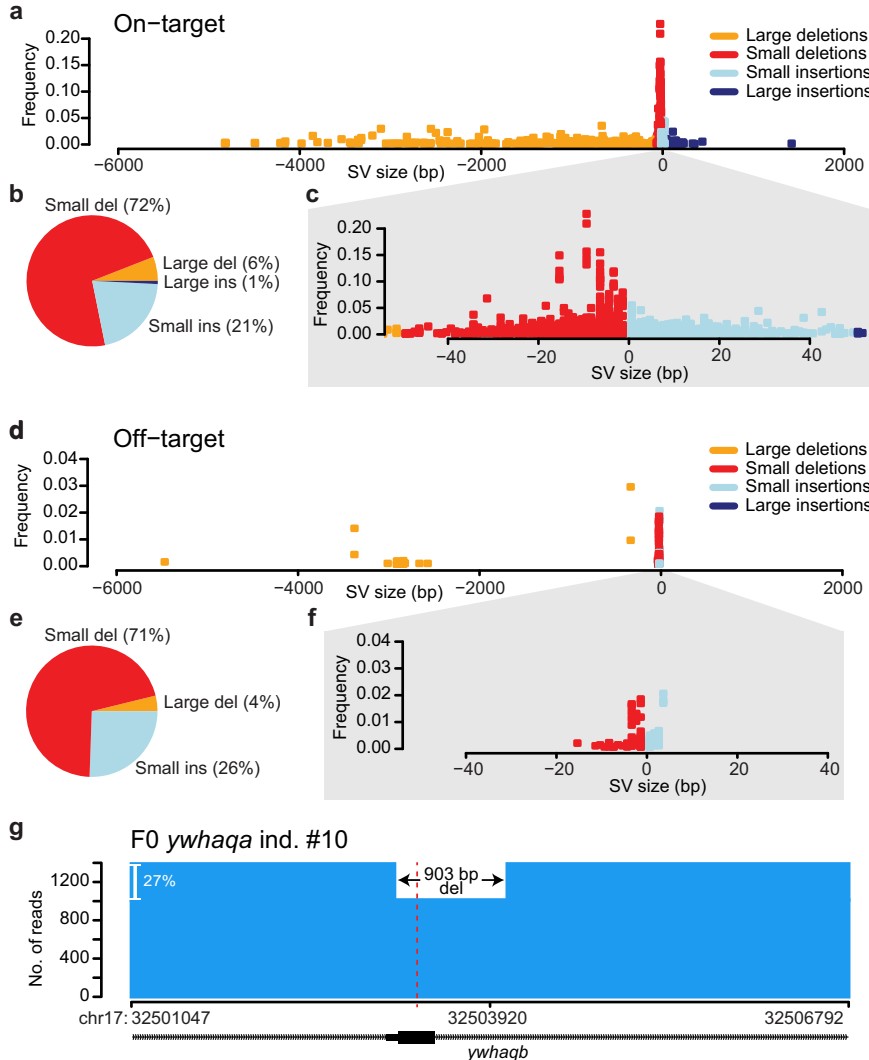

**Fig. 5 Size distribution of CRISPR-Cas9-induced mutations. a** A dot plot showing the size distribution of Cas9-induced on-target mutation events (large and small insertions and deletions) in pools of founder larvae. Each point in the graph visualizes a specific event in a sample, where the *x*-axis displays the size of the variant (negative values for deletions and positive values for insertions), and the *y*-axis displays the frequency of the variant in the individual sample. **b** Fractions of small and large insertions and deletions at on-target sites in pools of founder larvae. **c** Zoomed-in view of the plot in (**a**), only visualizing small insertions and deletions. **d** Size distribution of Cas9-induced off-target mutations (*sh2b3* off-target 1, *ywhaqa* off-target 1 and 2) in pools of founder larvae. **e** Fractions of small and large insertions and deletions at the three off-target sites in pools of founder larvae. **f** Zoomed-in view of the plot in d), only visualizing small insertions and deletions. **g** An example of an adult founder fish with a 903 bp deletion at *ywhaqa*'s off-target 2 that spans an entire exon of *ywhaqb*. The coverage plot shows the number of reads with the 903 bp deletion and the number of reads that lack the deletion (i.e., unmodified and other Cas9-induced variants). The Cas9 cleavage site is indicated by the dashed red line. Source data are provided as a Source Data file.

introduced by DNA amplification, long-range sequencing, or the downstream analysis.

In the founder larvae, about 7% of the editing outcomes at on-target sites correspond to insertions or deletions of ≥50 bp. At the off-target sites, this percentage is slightly lower (4%), but also more uncertain due to the small number of off-targets observed in total. Based on our results, we have no reason to presume that the frequency of SVs is significantly different between on- and off-targets, but rather that DNA repair through end-joining may introduce such events at a low probability at all sites undergoing Cas9 cleavage. Therefore, based on our results, we estimate the abundance of SVs during early cell division at 6% i.e., the combined percentage of SVs at both on- and off-targets. At the on-target sites, several large SV events can also be observed in adult founders, whereafter they in some instances segregate to the next generation. Four of the 46 juvenile F1 fish we examined carried a large deletion or a large insertion at the on-target site. We also

observed a 903 bp deletion at an off-target site that removes an exon of *ywhaqb* in one F0 individual. Unwanted large SVs in coding regions are problematic, but even more so when they occur at off-target sites, where they likely remain undetected.

Our data also point to several unexpected features of the CRISPR-Cas9 system that warrant further investigation. Firstly, we find that the germ cells of founder fish are mosaic. This is an important finding that can increase our understanding of how CRISPR-Cas9 events are inherited to the next generation. Moreover, 15% of the juvenile F1 fish seem to be homozygous for one specific CRISPR-Cas9 editing event, while the remaining ones are compound heterozygous. For the two F1 individuals undergoing nanopore WGS, the allelic imbalance could be explained by a dinucleotide repeat insertion preventing adequate amplification of the affected allele by long-range PCR. In other cases, the homozygosity may be caused by larger events induced by CRISPR-Cas9 that result in failed amplification of the affected

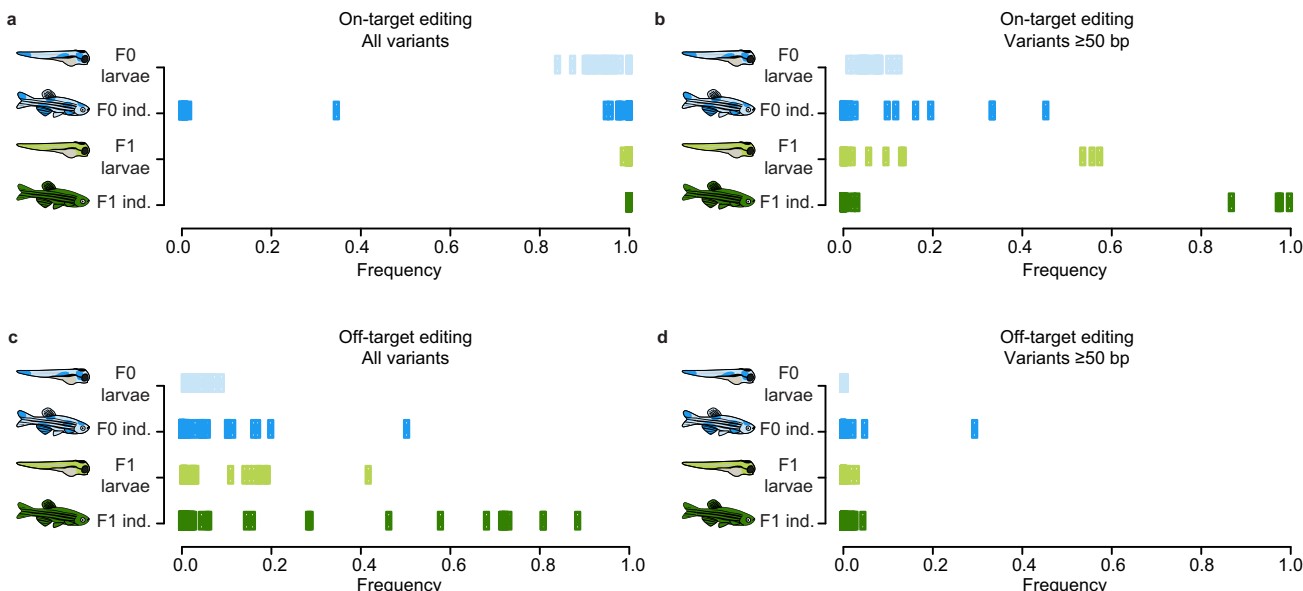

**Fig. 6 Summary of Cas9-induced variants at on- and off-target sites.** The frequencies of Cas9-induced variants in all analyzed samples for the F0 and F1 generations are displayed in the summary plots (**a**–**d**). Each point represents one sample (i.e., a pool of larvae or an individual juvenile/adult fish). **a** The total frequencies of Cas9-induced variants at on-target sites. **b** The frequencies of SVs induced by Cas9 at on-target sites. **c** The total frequencies of Cas9-induced variants at off-target sites (*sh2b3* off-target 1, *ywhaqa* off-targets 1 and 2). **d** The frequencies of SVs induced by Cas9 at off-target sites. Source data are provided as a Source Data file.

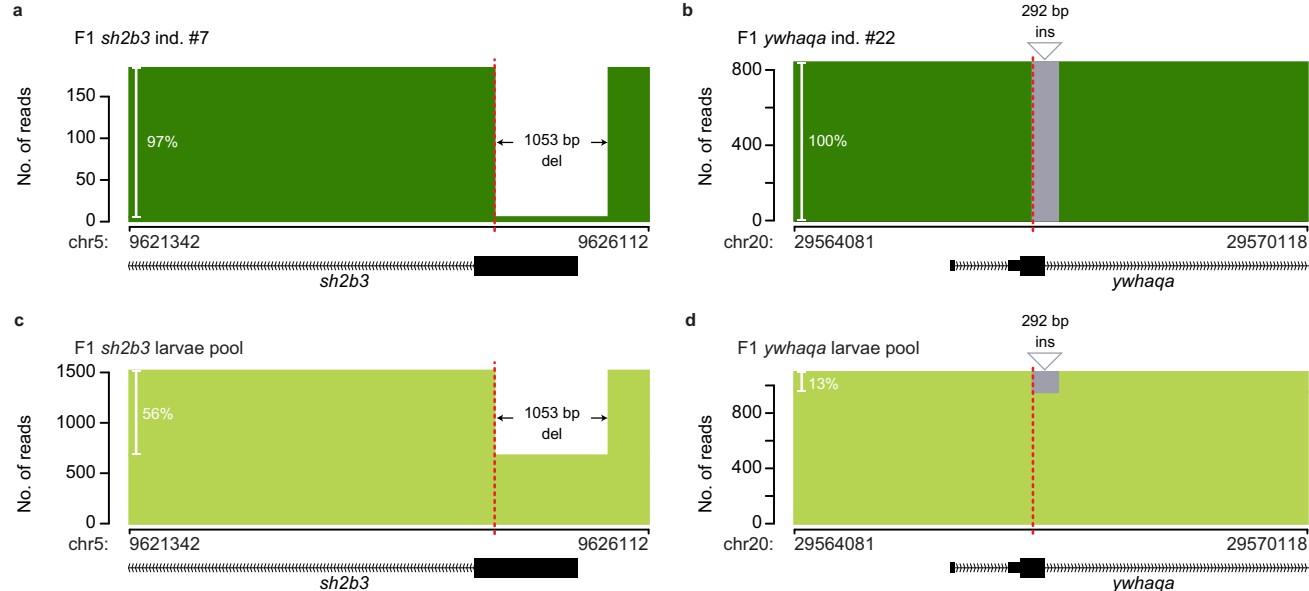

**Fig. 7 Large structural variants in individual F1 zebrafish.** Examples of large SVs at on-target sites in individual juvenile F1 zebrafish, a 1053 bp deletion in *sh2b3* (**a**) and a 292 bp insertion in *ywhaqa* (**b**). **c**, **d** The same variants are observed in pools of F1 larvae from the same F0 pair as the individuals in (**a**) and (**b**). The plots show the number of reads with the large SV and the reads without the variant (i.e., unmodified and other Cas9-induced variants). The Cas9 cleavage sites are indicated by the dashed red lines.

allele, loss-of-heterozygosity within the region, or alternatively, that the homozygosity is driven by non-random DNA repair resulting in the same mutations across multiple founders[40]. In any case, based on our results, more experiments are required to further improve our understanding of the multigenerational genomic consequences of CRISPR-Cas9 editing.

Even though this work is based on DNA from a large number of individuals, and using modern genomics technologies, many factors and parameters in the experiment could have influenced the results. For example, we microinjected RNPs to get the CRISPR-Cas9 machinery into the cells. Different results might have been obtained if we had used other methods for transfection. Additional factors such as differences in DNA repair system between cell types, or the total concentration of Cas9 within the cells, could also influence the editing efficiency and outcomes at the on- and off-target sites. For these reasons, it would be desirable to perform similar experiments also in other sample types and organisms. A further limitation of our study is that long-range PCR is unable to capture large genome aberrations and complex genomic rearrangements, such as chromothripsis[21]

and whole chromosome deletions[17,20]. Detecting such events requires an additional analysis of the edited samples, for example through long-read WGS as performed here, or by an alternative method for targeted long-read sequencing[42–45]. However, since we require viable outcome for the zebrafish, it is unlikely that large and complex genome alterations will be heritable and present in the analyzed samples.

Our finding that CRISPR-Cas9 can induce large SVs at on- and off-target sites in vivo does not mean we should stop using this powerful tool. For genetic screens in cellular systems or for functional experiments in model organisms, the impact of these large SV events will be relatively modest, since only a limited number of individuals or samples are likely to be affected. However, for clinical applications, such as genome editing in monogenetic disorders, it is critical to identify potentially serious adverse effects caused by a priori unexpected genome editing in the cells of interest. Lastly, when it comes to the manipulation of human embryos, our study adds yet more arguments for caution, due to the unintended mutations that can have consequences for the individual and, in some cases, future generations.

In conclusion, by applying new genomics tools and carefully designed experiments, we can learn more about the consequences of CRISPR-Cas9 editing in vivo, while at the same time developing improved strategies to validate edited cells. Based on our findings, we propose the following three-step approach to verify CRISPR-Cas9 genome editing outcomes for clinical applications: (i) employ an in vitro-method, such as Nano-OTS, to detect where Cas9 cleavage sites are located in DNA from the individual and—if possible—in (a) cell type(s) of interest, (ii) perform long-read re-sequencing of the on-target and predicted off-target sites, to determine the genotypes in each individual sample, and (iii) use an orthogonal method like CAST-seq[46], HTGTS[28], indirect sequence capture[47], amplification-free long-read enrichment[42,44,45], or whole-genome sequencing, to identify large and complex CRISPR-induced genome aberrations that may otherwise remain unnoticed. The last step is of particular relevance for samples with suspected allelic dropout. However, if long-read technology development continues on the current trajectory of increased throughput, improved accuracy, and reduced cost, it might soon be feasible to replace steps (ii) and (iii) by long-read whole-genome sequencing at high coverage. This would enable the detection of all possible types of CRISPR-Cas9 induced events from a single sequencing run without a need for primer design and targeted assays. Efficient validation methods that enable detection of small indels, larger SVs, and other unexpected genome editing outcomes, will be an important step towards a safer use of CRISPR-Cas9 for therapeutic purposes.

## Methods

**Zebrafish handling and CRISPR-Cas9 genome editing.** All zebrafish experiments and husbandry were conducted in accordance with Swedish and European regulations, and have been approved by the Uppsala University Ethical Committee for Animal Research (Dnr 5.8.18-13680/2020). The genes of interest were targeted using one gRNA per orthologue that had an anticipated efficiency >90%. RNA duplexes of the chemically synthesized Alt-R® crRNA (IDT) and Alt-R® tracrRNA (IDT) were complexed with Alt-R® S.p. Cas9 nuclease, v.3 (IDT) to form "duplex guide RNPs" (dgRNPs), as described by Hoshijima K et al.[39]. The dgRNPs were then injected into fertilized zebrafish eggs at the 1-cell stage. Uninjected embryos were kept and used as controls. The injected founder embryos were raised to adulthood at which time random mating pairs were in-crossed. Pools of 25–30 five- or ten-day-old F0 larvae, individual F0 adult fish, pools of 30 five-day-old F1 larvae, and fin clips individual F1 fish were collected throughout the experiment for downstream analyses. The complete sample collection is described in Supplementary Table S3. Adult zebrafish were sacrificed by prolonged exposure to tricaine, followed by snap freezing in liquid nitrogen to ensure DNA integrity.

**Extraction of genomic DNA from zebrafish.** All samples were extracted using the MagAttract HMW DNA Kit (Qiagen) and the "Manual Purification of High-Molecular-Weight Genomic DNA from Fresh or Frozen Tissue" protocol

according to the manufacturer's instructions. A tissue homogenization step using a pestle was added to the protocol prior to the lysis step. DNA integrity of the extracted samples was assessed using the Femto Pulse system (Agilent Technologies) using the Genomic DNA 165 kb kit.

**Detection of off-target sites using Nano-OTS.** Genomic DNA was sheared to 20 kb fragments using the Megaruptor 2 (Diagenode) and size selected with a 10 kb cut-off using the BluePippin system (Sage Science). 4 µg of sheared and size-selected DNA was then used for Nano-OTS library preparation as described by Höijer et al.[33]. A detailed description of all the steps of the Nano-OTS protocol is available from protocols.io (https://www.protocols.io/view/nano-ots-bp5smq6e). The sequences for all 23 gRNAs investigated by Nano-OTS are available in Supplementary Table S1. To increase coverage, two separate libraries were prepared and sequenced on one R9.4.1 flow cell each. Guppy v4.0 was used for base calling.

**Alignment of reads and detection of off-target sites.** The reads from Nano-OTS were aligned to the GRCz11 reference genome using minimap2[48], after which the Cas9 cleavage sites were predicted using v1.8 of the Insider software (https://github.com/UppsalaGenomeCenter/InSiDeR). For each predicted Cas9 cleavage site, the corresponding sequence from GRCz11 was extracted in a +−40 bp window surrounding the Cas9 cleavage site. All sequences containing gaps (N's) were filtered out since we were only interested in detecting gRNA binding events in high-quality regions of the zebrafish genome. For the remaining sequences, we globally aligned against all gRNA sequences using v6.6.0 of EMBOSS-Needle with default settings[49]. Only sequences with an alignment score of >55 to a certain gRNA were considered positive binding sites.

**Amplicon construction and sequencing.** Primers were designed for all on-target and off-target sites predicted by Nano-OTS for the four gRNAs. Primers for three off-target sites for the *nbeal2* gRNA were excluded due to PCR optimization difficulties or because of issues with the GRCz11 zebrafish reference genome. The amplicons range from 2.6 to 7.7 kb in size. Primer sequences, expected amplicon sizes, and primer coordinates can be found in Supplementary Table S4. Long-range PCRs were performed using the PrimeStar GLX Polymerase (Takara Bio) according to the manufacturer's instructions, using either the standard or two-step PCR cycling protocol. 0.2 µg/µl bovine serum albumin (BSA) was added to the PCR reactions for improved performance. PCRs were performed using 30 ng of genomic DNA from pooled or individual zebrafish DNA extractions. Amplicons originating from different primer pairs were pooled in an equimolar fashion. Amplicon pools were barcoded and sequenced on PacBio's Sequel system using the SMRTbell® Express Template Prep Kit 2.0 and the PacBio Barcoded Overhang Adapter Kit 8A and 8B for SMRTbell construction, and 3.0 sequencing and binding chemistry using a 10 h movie time.

**Analysis of on- and off-target mutations in long amplicon data.** CCS reads for the long amplicons sequenced on PacBio's Sequel system were generated using SMRTLink v10.1, after which alignment was performed to GRCz11 using minimap2[48]. On- and off-target editing efficiencies were calculated as the fraction of reads containing insertions and deletions at the Cas9 digestion site. To ensure that indel variation at the Cas9 cleavage sites is caused by CRISPR-Cas9 genome editing, and not by genetic variation in the zebrafish, systematic errors introduced in the sequencing, or alignment artefacts, we removed all sites having a frequency of at least 0.5% indel mutations in the control samples ($F0_{wt}$ pools). After alignment, all reads covering a specific on-target or off-target site were analyzed using the software SIQ (https://github.com/RobinVanSchendel/SIQ). SIQ performs a detailed analysis of all reads covering a specific target and reports the identified editing events along with their frequencies. To remove potential false positive events reported by SIQ in the edited samples, all events detected in the control samples were flagged and considered as unedited. Custom R scripts were used to visualize the SIQ results. In cases where an SV event simultaneously contain an insertion and a deletion, the event was visualized either as an insertion or as a deletion depending on which part of the SV had the largest size.

**Nanopore whole genome sequencing and downstream analysis.** Nanopore WGS was performed to verify the genome editing outcomes in two juvenile *ywhaqa*-edited F1 zebrafish (individuals #5 and #17). In addition, two control samples were subject to nanopore whole-genome sequencing: one juvenile *sh2b3* F1 zebrafish, and a *ywhaqa* F0 pool of larvae. For each F1 fish to be sequenced, one microgram of genomic DNA extracted from a fin clip. The DNA was then used for sequencing library preparation, following the protocol for sequencing of genomic DNA by ligation using the SQK-LSK110 kit (Oxford Nanopore Technologies). Each library was sequenced for 72 h on one R9.4.1 flow cell using the MinION system. Due to high molecular integrity, one of the samples was sheared to 20 kb fragments using Megaruptor 3 (Diagenode) at speed 30 prior to library preparation. Guppy v5.0 was used for base calling. SV calling was performed on all samples using NGMLR v.0.2.7 for alignment to the GRCz11 reference genome and Sniffles v.1.0.12[50]. Genomic DNA from 30 pooled F0 larvae was subject to the Short-read eliminator kit protocol (Circulomics), to remove DNA fragments <25 kb. Size selected DNA (2.5 µg) was then used for library preparation using the protocol for sequencing of genomic DNA by ligation using the SQK-LSK110 kit (Oxford Nanopore Technologies). Half of the library was loaded on one PromethION

flow cell and sequenced for 22 h before pausing the sequencing run and washing the flow cell using the EXP-WSH004 Flow cell wash kit (Oxford Nanopore Technologies). The remaining sequencing library was then loaded and sequenced for a total run time of 72 h, followed by Guppy base calling. The CNV analysis was performed individually for each of the four samples, by aligning the nanopore reads using minimap2[48] and subsequently analyzing the coverage depth in 50 kb windows across the GRCz11 reference using the bamCoverage tool available from v3.3.2 of deepTools[51].

**Reporting summary**. Further information on research design is available in the Nature Research Reporting Summary linked to this article.

## Data availability

The sequence data that support the findings of this study have been deposited in the NCBI Sequence Read Archive (SRA) with the accession code PRJNA772901. Source data are provided with this paper. The GRCz11 reference genome used in this study is available in the file "danRer11.fa.gz" from https://hgdownload.soe.ucsc.edu/goldenPath/danRer11/bigZips/. Source data are provided with this paper.

## Code availability

The Insider tool for identification of CRISPR-Cas9 off-target sites is available from GitHub (https://github.com/UppsalaGenomeCenter/InSiDeR)[52]. The SIQ software for analysis of editing outcomes is available from GitHub (https://github.com/RobinVanSchendel/SIQ)[53].

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

## Acknowledgements
Sequencing was performed by the SciLifeLab National Genomics Infrastructure (NGI) in Uppsala, Sweden. Computations were performed on resources provided by SNIC through Uppsala Multidisciplinary Center for Advanced Computational Science (UPPMAX) under Project SNIC 2021/23-81. We would like to thank Mai-Britt Mosbech, Uppsala University, for advice and recommendations for high molecular weight DNA extractions in zebrafish. This study was supported by grants from Swedish Cancer Foundation and Swedish Research Council (U.G.). M.d.H. is supported by grants from the Swedish Heart-Lung Foundation (20200781, 20200602), the Kjell and Märta Beijer Foundation, and the Swedish Research Council (2019-01417).

## Author contributions
I.H., L.F., U.G., M.d.H., and A.A. conceived the study. I.H. and A.A. drafted the manuscript. I.H., A.E., and R.Ö. performed the experiments. I.H., R.v.S., S.B., M.T., and A.A. developed bioinformatics tools and performed the analyses. All authors read and approved the final manuscript.

## Funding

## Competing interests
The authors declare no competing interests.
