## [Peer Review File · Nature Communications]

Reviewers' Comments:

Reviewer #1:

Remarks to the Author:

This manuscript, by Hoijer et al., aims to assess unintended genome editing outcomes at both on- and off-target sites in vivo in zebrafish. Although gene editing is generally meant to induce small indels, recent literature has shown that it is also capable of inducing large deletions, truncations, and structural variants (SVs) at the target site. To date, studies of these unintended outcomes have only been performed in cell lines or embryos that were not brought to term. Moreover, with the exception of Zuccaro et al., no study has rigorously assayed whether these large SVs might also occur at off-target editing sites. Hoijer et al., laudably address these questions by long-read sequencing of PCR products in embryonic-edited zebrafish larvae, adults, and progeny of these edited fish. They find that both on- and off-target sites are susceptible to SVs at that CRISPR/Cas9 cut-sites, and that these unintended outcomes of editing are passed on to the next generation. Furthermore, they also show high rates of mosaicism of these unintended edits within individual fish.

The major shortcoming of the manuscript lies in the approach used to detect SVs, however this shortcoming is appropriately discussed and qualified in the text. By using large PCR amplicons, it is likely that some SVs that were larger than the amplicon were missed. This possibility is supported by the fact that some edits appeared homozygous. However, this shortcoming means the results are likely only underestimates of the true frequency of unintended SV, and thus further highlight the importance of the result. With only some minor clarifications of the text that would not require further review, I recommend this paper for publication in Nature Communications.

Specific points:

(1) The paragraph highlighting the data contained in Figure 6 requires clarification. Why are different thresholds used in the text for number of fish with off-target editing (20% of reads vs 10% of reads)? Also, this section should be clarified as to what data are actually being presented. For example, is this 12 of 46 fish with at least 1 off-target event present? When off-target editing is observed in 20% of reads, does this mean 20% of all sequencing reads at the three cut sites? Because these are F1 fish you would expect read counts to be ~0%, ~50%, or ~100% edited at each cut site, and so presenting the data as a summary statistic is not intuitive.

(2) The manuscript would benefit from some discussion of the difference in the prevalence of SVs between on- and off- target sites. This is most likely due to a lower editing rate at off-target sites, but is the ratio of indel:SV the same for both on- and off-target effects? Knowing this might contribute to some mechanistic understanding of how SVs arise.

(3) The manuscript should acknowledge that Zuccaro et al. did previously find large truncations at off-target sites.

Reviewer #2:

Remarks to the Author:

CRISPR-Cas9 induces large structural variants at on-target and off-target sites in vivo that segregate across generations

Hoijer et al

Review:

The study of Hoijer et al took advantage of the zebrafish system to examine the inheritance of CRISPR-Cas9 induced mutations from the founder F0 population to the F1 progeny. The strength of the zebrafish system is that the F1 progeny can be tested on an individual basis such that distinct modified alleles can be identified. The authors treated the F0 population with guide RNAs designed against four genes that have orthologues in human and utilized Long-Read sequencing of

long amplicons to identify CRISPR – induced mutations, with the emphasis on structural variations (SVs), such as long insertions and deletions. Overall, the authors provided a very accurate description of modified alleles and the quantitative assessment of large indels abundance in the F1 population and clearly demonstrated that large SVs and off-target mutations can be introduced in vivo and passed through the germline to the F1 generation. While the manuscript is concise and very well written, further analysis will strength the paper as suggested below:

1. The authors narrowed the scope of SVs to large indels. However, other structural variations such as translocations and other chromosomal aberrations were overlooked. The recent FDA hold-back on Allogene's allogeneic CAR-T therapy following chromosomal aberration is an example of the critical importance of detecting chromosomal aberrations. Particularly, in the “homozygous” mutants where the authors were unable to identify the second allele, FISH with probes placed proximally and distally to the cut site and some alternative primer pairs might give a hint on whether the chromosomal loss or other aberration had been involved.

2. There was no phenotypic analysis in the manuscript. Although the authors did mention that only live progeny was examined so that the potentially very harmful or lethal mutations or SVs would not be included in the F1 testing. At least, the identified SVs which impacted exons might demand more through phenotypic examination.

Furthermore, in the discussion, the authors proposed three-step approach to verify CRISPR-Cas genome editing outcomes for clinical applications however, they ignore some important genetic variants and SVs events that can not be characterized by the proposed long-read re-sequencing and therefore must be complemented by additional methods. These methods should include methods such as CAST-seq or HTGTS, among others, to detect other chromosomal aberrations.

***** General comments: We would like to thank the two reviewers for their evaluation of our work. Based on their insightful comments, we have now included additional data and made changes that has improved the quality of the manuscript. Please see below for point-by-point answers to all the specific questions and remarks. Throughout the manuscript text, all changes have been marked in red. *****

REVIEWER COMMENTS

Reviewer #1 (Remarks to the Author):

This manuscript, by Hoijer et al., aims to assess unintended genome editing outcomes at both on- and off-target sites in vivo in zebrafish. Although gene editing is generally meant to induce small indels, recent literature has shown that it is also capable of inducing large deletions, truncations, and structural variants (SVs) at the target site. To date, studies of these unintended outcomes have only been performed in cell lines or embryos that were not brought to term. Moreover, with the exception of Zuccaro et al., no study has rigorously assayed whether these large SVs might also occur at off-target editing sites. Hoijer et al., laudably address these questions by long-read sequencing of PCR products in embryonic-edited zebrafish larvae, adults, and progeny of these edited fish. They find that both on- and off-target sites are susceptible to SVs at that CRISPR/Cas9 cut-sites, and that these unintended outcomes of editing are passed on to the next generation. Furthermore, they also show high rates of mosaicism of these unintended edits within individual fish.

The major shortcoming of the manuscript lies in the approach used to detect SVs, however this shortcoming is appropriately discussed and qualified in the text. By using large PCR amplicons, it is likely that some SVs that were larger than the amplicon were missed. This possibility is supported by the fact that some edits appeared homozygous. However, this shortcoming means the results are likely only underestimates of the true frequency of unintended SV, and thus further highlight the importance of the result. With only some minor clarifications of the text that would not require further review, I recommend this paper for publication in Nature Communications.

***** Reply: We completely agree that long amplicon sequencing has important limitations when it comes to detection of SVs, and the same concern was also raised by Reviewer 2. To address this issue, we have now included data from whole genome long-read sequencing of two of the homozygous fish from the F1 generation. Please see the reply to Reviewer 2 below for more details on the whole genome sequencing experiments and outcomes. *****

Specific points:

(1) The paragraph highlighting the data contained in Figure 6 requires clarification. Why are different thresholds used in the text for number of fish with off-target editing (20% of reads vs 10% of reads)? Also, this section should be clarified as to what data are actually being presented. For example, is this 12 of 46 fish with at least 1 off-target event present? When off-target editing is observed in 20% of reads, does this mean 20% of all sequencing reads at the three cut sites? Because these are F1 fish you would expect read counts to be ~0%, ~50%, or ~100% edited at each cut site, and so presenting the data as a summary statistic is not intuitive.

***** Reply: These questions are very relevant, and we agree that the paragraph needs clarification. Our rationale for having different cut-offs for F0's and F1's is that the F0 fish are highly mosaic, while the F1 fish are either heterozygous or homozygous. For the mosaic F0 fish, we selected 10% as a cut-off since this reflects a moderate level of genome editing. Although we don't know anything about the mutational distribution of the germ cells, F0 individuals with at least 10% editing should have a decent chance to pass on CRISPR-induced mutations to their offspring. For the F1 individuals, we selected a more stringent cut-off, since as many as 50% of the reads are expected to be edited for a heterozygous individual. Due to allelic bias and random effects, the allelic distribution of a heterozygous individual might be quite**

different from 50%, and we therefore put the threshold for the F1's at 20%. However, we understand that the cut-offs can be seen as somewhat arbitrary. In the revised manuscript, we acknowledge the difficulties in doing this type of comparisons between the F0's and F1's and explain our rationale for choosing the 10% vs 20% thresholds.

Regarding off targets, the reviewer is correct in assuming that we meant that 12 of 46 fish have at least one off-target site present at 20%. We have now clarified this in the text and would like to thank the reviewer for pointing this out."

(2) The manuscript would benefit from some discussion of the difference in the prevalence of SVs between on- and off- target sites. This is most likely due to a lower editing rate at off-target sites, but is the ratio of indel:SV the same for both on- and off-target effects? Knowing this might contribute to some mechanistic understanding of how SVs arise.

***** Reply: Yes, this indeed is an interesting discussion topic. The prevalence of SVs is slightly higher at the on-target sites (7%) as compare to off-target sites (4%), and the combined prevalence at on- and off-target sites is 6%. In the abstract, we now write that the prevalence of SVs in our CRISPR experiments is 6%, instead of 7% as before, since this is a better estimate for the complete dataset including both on- and off-target sites. Just as the reviewer points out, the small discrepancies between on- and off-targets are likely explained by the low off-target editing rate, leading to few observations of SVs and uncertain statistics. The above percentages and our interpretation of the results are now mentioned in the second paragraph of the Discussion. *****

(3) The manuscript should acknowledge that Zuccaro et al. did previously find large truncations at off-target sites.

***** Reply: We now specifically acknowledge the work by Zuccaro et al in the Introduction. *****

Reviewer #2 (Remarks to the Author):

CRISPR-Cas9 induces large structural variants at on-target and off-target sites in vivo that segregate across generations

Hoijer et al

Review:

The study of Hoijer et al took advantage of the zebrafish system to examine the inheritance of CRISPR-Cas9 induced mutations from the founder F0 population to the F1 progeny. The strength of the zebrafish system is that the F1 progeny can be tested on an individual basis such that distinct modified alleles can be identified. The authors treated the F0 population with guide RNAs designed against four genes that have orthologues in human and utilized Long-Read sequencing of long amplicons to identify CRISPR – induced mutations, with the emphasis on structural variations (SVs), such as long insertions and deletions. Overall, the authors provided a very accurate description of modified alleles and the quantitative assessment of large indels abundance in the F1 population and clearly demonstrated that large SVs and off-target mutations can be introduced in vivo and passed through the germline to the F1 generation. While the manuscript is concise and very well written, further analysis will strength the paper as suggested below:

1. The authors narrowed the scope of SVs to large indels. However, other structural variations such as translocations and other chromosomal aberrations were overlooked. The recent FDA hold-back on Allogene's allogeneic CAR-T therapy following chromosomal aberration is an example of the critical importance of detecting chromosomal aberrations. Particularly, in the "homozygous" mutants where the authors were unable to identify the second allele, FISH with probes placed proximally and distally to the

cut site and some alternative primer pairs might give a hint on whether the chromosomal loss or other aberration had been involved.

***** Reply: This is a very important point, which was also raised by Reviewer 1 above. To understand whether there are any larger SVs, outside of the detection limit of long-amplicon sequencing, we performed nanopore-based whole genome sequencing (WGS) for two F1 fish. One of these F1 fish was considered “homozygous”, while the other one contained two alleles but with a very skewed allele distribution. Interestingly, the WGS experiment revealed that both of these F1 are in fact heterozygous. Furthermore, both samples carried an AT-repeat insertion, rendering one of the alleles difficult to amplify by long-range PCR. However, there was no evidence for any other inversions, translocations, whole chromosome deletions, or other types of complex SVs in the investigated individuals. Our WGS experiment thus shows that long-amplicon sequencing can give biased results in repetitive regions, and supports the argument to perform additional validation with an orthogonal method (see also point below). Ideally, the orthogonal validation should be performed by an amplification-free sequencing protocol to avoid bias and allelic dropout.**

In conclusion, the nanopore-based WGS experiment highlights some unexpected shortcomings of our long-amplicon sequencing assay. But nevertheless, we find the results of this new analysis to be highly interesting and of importance for how to best validate CRISPR editing outcomes. In the Results section, we have added a subsection on the nanopore WGS analysis. We have also made some minor changes to the Discussion to put our WGS results into context. There is also a section in the Methods describing how the WGS experiments were performed. ***

2. There was no phenotypic analysis in the manuscript. Although the authors did mention that only live progeny was examined so that the potentially very harmful or lethal mutations or SVs would not be included in the F1 testing. At least, the identified SVs which impacted exons might demand more through phenotypic examination.

***** Reply: The reviewer raises an interesting point that we also contemplated implementing. However, examining the association of large structural variants with relevant phenotypes is currently not feasible. The genes presented in the present study were previously selected as candidates for a role in cardiometabolic diseases and their risk factors. For these diseases, we have developed image and CRISPR-Cas9-based model systems in which F1 or F0 10-day old zebrafish larvae are phenotypically characterised, followed by paired-end sequencing at the CRISPR-targeted sites (2x250 bp, in F1 larvae) or fragment length PCR analyses (in F0 larvae), in DNA obtained from individual larvae. To examine the role of candidate genes in cardiometabolic diseases and their risk factors, associations of relevant phenotypes and Cas 9-induced mutations are examined in single larvae (for more information, see www.biorxiv.org/content/10.1101/502674v3; <https://pubmed.ncbi.nlm.nih.gov/32678143/>). However, for the PCR-free, long-read sequencing protocols used in the present study, pooling of larvae is required to obtain sufficient DNA that is of high enough quality. In adult fish on the other hand, sufficient DNA can be obtained for long-read sequencing in individual fish, but validated phenotyping pipelines for relevant traits are not available. In addition, developing and validating model systems in adult fish is beyond the scope of the present study. Importantly, while our results have important implications for guidelines in clinical applications, our results suggest that large on and off-target structural variants are unlikely to influence the results of systematic CRISPR-based genetic screens, with a low proportion of gRNAs being affected in a low proportion of samples. Hence, to examine the role of these genes in cardiometabolic diseases, we can simply focus on the results of ongoing screening efforts that in due time will be published separately. *****

Furthermore, in the discussion, the authors proposed three-step approach to verify CRISPR-Cas genome editing outcomes for clinical applications however, they ignore some important genetic variants and SVs events that can not be characterized by the proposed long-read re-sequencing and therefore must be complemented by additional methods. These methods should include methods such as CAST-seq or HTGTS, among others, to detect other chromosomal aberrations.

***** Reply: We agree that our proposed strategy for validation of genome editing outcomes was unsatisfactory. As the reviewer points out, complex genetic variation and other genomic aberrations might be missed. For clinical implementation, it is crucial that also such events can be detected through robust and sensitive methods. We have therefore rewritten the last paragraph of the Discussion. In the new paragraph, we highlight methods like CAST-seq, HTGTS, and others, as important complements to the long-amplicon method used in this study. We also mention long-read WGS as a potential future approach. Even though long-read WGS might not be feasible today, due to high cost and insufficient throughput, it would enable detection of all variants in a single amplification-free assay, without the need to design PCR primers for the on- and off-target regions. It is crucial to use the best methods available for clinical CRISPR validation assays, since it ultimately affects patient safety. We thank the reviewer for raising this important issue, and believe that the new text better describes how the methods developed by us and others can complement each other towards a safer use of CRISPR in the clinic. *****

Reviewers' Comments:

Reviewer #1:

Remarks to the Author:

The revised version of this manuscript by Höijer et al., clarifies earlier presentations of the data and adds appropriate discussion. Commendably, the authors also undertook long-read whole genome sequencing of edited zebrafish and their progeny in order to overcome the shortcoming of their earlier PCR based approach. Although this was only performed in a few samples, it greatly adds to the manuscript. The new sequencing data highlights that in addition to large deletions and structural rearrangements, hard to amplify DNA may disrupt PCR-based analyses, yielding false interpretations of homozygosity after editing.

I enthusiastically recommend this manuscript for publication in Nature Communications but make some minor points that I do not need to further review:

- (1) I still do not find it intuitive to pool all the off-target sites from a single sample into a single summary statistic per sample in Figure 6. Nevertheless, this is a fairly minor point of presentation, and leave it to the author's discretion.
- (2) Lines 394-395 of the new methods section should be clarified to say "For each F1 fish to be sequenced, one microgram of genomic DNA extracted from a fin clip. The DNA was then used for sequencing library preparation...". The previous version made it seem like three individual fish were sequenced for each sample.
- (3) The new Figure S1 would benefit from adding coordinates to show the repeat location relative to the target site in a graphical manner.
- (4) The bin sizes should be noted in the legend plots in the new Figure S2.
- (5) Line 223 change "cannot" to "did not"
- (6) Line 51 – change to "Long-read sequencing suffers less from these limitations from these limitations". As noted in your manuscript, long-range PCR preceding long-read sequencing can still be problematic for variant detection.

Reviewer #2:

Remarks to the Author:

The authors have taken steps to address all my comments. The manuscript should be accepted for publication.

***** We again would like to thank the reviewers for their continued evaluation and encouraging comments. Please see below for point-by-point answers to the specific queries. Throughout the manuscript text, all changes have been marked in red. *****

REVIEWERS' COMMENTS

Reviewer #1 (Remarks to the Author):

The revised version of this manuscript by Höjjer et al., clarifies earlier presentations of the data and adds appropriate discussion. Commendably, the authors also undertook long-read whole genome sequencing of edited zebrafish and their progeny in order to overcome the shortcoming of their earlier PCR based approach. Although this was only performed in a few samples, it greatly adds to the manuscript. The new sequencing data highlights that in addition to large deletions and structural rearrangements, hard to amplify DNA may disrupt PCR-based analyses, yielding false interpretations of homozygosity after editing.

I enthusiastically recommend this manuscript for publication in Nature Communications but make some minor points that I do not need to further review:

(1) I still do not find it intuitive to pool all the off-target sites from a single sample into a single summary statistic per sample in Figure 6. Nevertheless, this is a fairly minor point of presentation, and leave it to the author's discretion.

***** Reply: Even though the results could have been presented separately for each off-target, we prefer to show an overview of combined data for the three off-target sites. We believe that gives a better high-level overview of the results. *****

(2) Lines 394-395 of the new methods section should be clarified to say "For each F1 fish to be sequenced, one microgram of genomic DNA extracted from a fin clip. The DNA was then used for sequencing library preparation...". The previous version made it seem like three individual fish were sequenced for each sample.

***** Reply: This sentence has been changed according to the reviewer's suggestion. *****

(3) The new Figure S1 would benefit from adding coordinates to show the repeat location relative to the target site in a graphical manner.

***** Reply: Figure S1 has been changed, and now includes the requested information. *****

(4) The bin sizes should be noted in the legend plots in the new Figure S2.

***** Reply: The bin sizes (50 kb) are now mentioned in the legend of Figure S2. *****

(5) Line 223 change "cannot" to "did not"

***** Reply: Ok, fixed. *****

(6) Line 51 – change to "Long-read sequencing suffers less from these limitations from these limitations". As noted in your manuscript, long-range PCR preceding long-read sequencing can still be problematic for variant detection.

***** Reply: Ok, fixed. *****

Reviewer #2 (Remarks to the Author):

The authors have taken steps to address all my comments. The manuscript should be accepted for publication.